# HIV testing uptake and yield among sexual partners of HIV-positive men who have sex with men in Zhejiang Province, China, 2014-2016: A cross-sectional pilot study of a choice-based partner tracing and testing package

**Mingyu Luo**[1], **Katrina Hann**[2], **Guomin Zhang**[3], **Xiaohong Pan**[1]\*, **Qiaoqin Ma**[1]\*, **Jun Jiang**[1], **Lin Chen**[1], **Shichang Xia**[1]

**1** Center for Disease Control and Prevention of Zhejiang Province, Hangzhou, China, **2** Sustainable Health Systems, Sierra Leone, **3** National Immunization Program, Chinese Center for Disease Control and Prevention, China

\* xhpan@cdc.zj.cn (XP); qqma@cdc.zj.cn (QM)

## Abstract

### Background

Measures to effectively expand tracing and testing to identify undiagnosed HIV infections are significant for the control of HIV/AIDS epidemic among men who have sex with men (MSM). We piloted a choice-based tracing and testing package aimed at improving partner tracing, uptake, and yield of HIV testing for sexual partners of newly diagnosed HIV-positive MSM.

### Methods

This package was piloted in the cities of Hangzhou and Ningbo, Zhejiang province, China from June 2014 to June 2016. The package adopted four modes: couples' HIV counseling and testing (CHCT), information assisted partner notification (IAPN), assisted HIV self-testing (HIVST) and patient referral. Data regarding sociodemographic factors and sexual behaviors between HIV-positive MSM and their sexual partners, as well as tracing and testing outcomes of each mode, were collected.

### Results

Among 2,495 newly diagnosed HIV-positive MSM, 446(18%) were enrolled as index cases (ICs) through two rounds of contact tracing. The ICs disclosed a total of 4,716 sexual partners, of whom 548 (12%) were reachable. The pilot study resulted in a testing uptake of 87% (478/548), and a yield of 16% (74/478) among sexual partners. The generalized linear mixed model showed that the odds of a reachable sexual partner enrolled via IAPN taking an HIV test were 290% greater than that of a partner traced via CHCT (95% CI: 1.6, 9.3).

**Data Availability Statement:** The data underlying this study cannot be publicly shared because the

database contains personal and sexual information of HIV positive persons, which is sensitive, confidential and under restrict management by Chinese Law. The anonymized data will be available upon request to the Zhejiang Provincial Center for Disease Control and Prevention Ethics Review board. Zhejiang Provincial Center for Disease Control and Prevention Ethics Review Board: Address: NO.3399, Binsheng Road, Binjiang District, Hangzhou City, Zhejiang Province, China Please contact: Mr. Zhenggang Jiang Email: zhgjiang@cdc.zj.cn, TEL:+86 571-87115105.

**Funding:** This study was supported by Key Project on Social Development among S&T Major Project of Zhejiang Province, China (2013C03047-1), Zhejiang Provincial Medicine Science and Technology Plan (2015PYA004), National Science and Technology Major Project of China (2017ZX10201101), The Training Project of Young Scientific and Technological Innovative Talents of Zhejiang Provincial Center of Disease Control and Prevention. The funders had no role in study design, data collection and analysis, decision to publish, or preparation of the manuscript.

**Competing interests:** The authors have declared that no competing interests exist.

## Conclusions

A choice-based tracing and testing package can feasibly expand HIV testing uptake and case finding among sexual partners of HIV-positive MSM. IAPN may be an acceptable option to reach sexual partners for whom limited contact information is available.

## Introduction

Men who have sex with men (MSM) have a substantial risk of contracting HIV/AIDS infections in China, where newly diagnosed infection increased almost six-fold between 2010 to 2016, and where the prevalence of HIV infections in MSM was 7.75% in 2016 (similar to European countries) [1,2]. Failure to detect an infection is a significant driver of the HIV epidemic among MSM [3,4]. Globally, HIV testing rates among MSM range from 50% to 70% [5–7], indicating a large proportion of undiagnosed HIV infections.

HIV tracing and testing approaches are essential in identifying undiagnosed infections and preventing further HIV transmission among sexual partners of HIV-positive MSM [8,9]. In 2003, the Chinese government began to enforce the free voluntary counselling and testing (VCT) policy [10], which resulted in the introduction of VCT clinics, among other services [11]. The introduction of VCTs in China allowed MSM to visit testing sites and determine their infection status through passive-case finding, in which MSM initiate HIV testing, even in areas with significant NGO outreach efforts [12]. Subsequently, the rate of HIV testing among MSM increased from 50% to 65% from 2011 to 2015 [1]. However, as 25% of HIV infections among MSM remain undiagnosed [13], active case-finding approaches in which healthcare providers target this key population may be a significant step in addressing this testing gap.

Emerging evidence highlights the acceptability, feasibility, and transferability of existing intervention models as key factors in active case finding and referral. The World Health Organization lists options for assisted partner referrals, in which trained health care providers assist clients in disclosing their partners' potential exposure to HIV infections and help refer their partners to HIV testing services (HTS).

In assisted partner notification services, information-based tracing and testing modes that utilize social network applications may provide adequate convenience and privacy protection for index cases by providing a direct line of communication line to health providers and their contacts [14,15]. By tracing sexual partners using social media account identifiers, contact between the HIV-positive MSM and their sexual partners can be avoided [16]. Social media point-to-point interventions, in which a healthcare provider communicates directly with the contact, have been shown to be feasible in reaching MSM in previous studies [14]. This approach is effective in identifying high numbers of HIV-positive individuals in need of treatment and is cost-effective in locating and contacting partners [17].

Couples' HIV counseling and testing (CHCT) is a subtype of assisted partner notification in which couples test, share results, and receive interventions together. This also may be an acceptable approach among MSM and their sexual partners [18,19]; however, more evidence on contact-tracing of HIV-positive MSM is needed.

Assisted HIV Self-Testing (HIVST) refers to individuals who are self-testing for HIV but receive an in-person demonstration from a trained provider or peer before or during HIVST. HIVST is an acceptable strategy that can increase testing uptake among high-risk populations that are less likely to have access to testing [17,20,21]. In HIVST, oral HIV self-testing may be a

complementary method for screening sexual partners who fear disclosure of HIV status, however, it has not consistently been found as acceptable [22,23].

In practice, active and passive contact tracing are complementary approaches: care providers may offer more than one testing mode for clients, which allows them to select an option that reduces each individual's specific barriers to testing. Sharma et al. [24] explored MSM attitudes and usage preference towards six different HIV testing modes and found that MSM of different sub-groups showed low acceptability for one intervention option and had specific intervention preferences [24].

In China, there is limited data on choice-based partner tracing and testing for sexual partners of HIV-positive MSM. This type of data is essential to HIV prevention and control programs, as it provides guidelines for active case-finding strategies targeting special populations such as MSM and their sexual partners. In this study, we aimed at analyzing the uptake and infection status of HIV testing for sexual partners of newly diagnosed HIV-positive MSM. We also describe the program's intervention components and outcomes in the Zhejiang Province of China.

## Materials and methods

### Selection and description of participants

We defined the study population as newly diagnosed HIV-positive MSM and their sexual partners, who agreed to participate in the partner tracing and HIV testing program at VCT clinic study sites in the cities of Hangzhou and Ningbo from June 2014 through June 2016. Inclusion criteria for index cases (ICs) included the following: male, age 18 years or older, having been newly diagnosed as HIV positive within one month, having disclosed male sexual partners (including oral intercourse or anal intercourse) to VCT staff in pre- or post-testing counseling at the time of diagnosis, and having agreed to participate in the partner tracing and testing program. We excluded any potential participants with a cognitive impairment, severe AIDS complication or other illnesses requiring hospitalization, or disabilities, as determined by staff at the VCT clinics. We defined inclusion criteria for sexual partners as individuals reported to be sexual partners of ICs in post-testing counseling at the time of HIV testing, but excluded those with cognitive impairments, severe AIDS complication or other illnesses requiring hospitalization, or disabilities, as determined by staff at VCT clinics. We defined reachable sexual partners as those who could be contacted by face-to-face, official landline/telephone, social software (such as QQ, WeChat and Blued) or other communication tools. We did not include an exclusion criterion for sexual partner age in order to identify any high social risk between adult IC and minor sexual partners. However, HIV-positive sexual partners aged under 18 years were interviewed with their legal guardians' written consent and presence.

### Technical information

**Study design.**    We used a cross-sectional design to describe a pilot choice-based partner tracing and testing program that offers a choice of four modes to analyze HIV testing uptake and identification of undiagnosed infections among sexual partners of HIV positive MSM.

**Study setting.**    The goal of the choice-based partner tracing and testing program was to increase HIV testing uptake and yield among sexual partners of newly diagnosed HIV positive MSM, by offering choices in a partner tracing and testing package. This package targeted sexual partners of newly diagnosed HIV positive MSM, as disclosed in post-testing counseling at VCT clinics. The staff from VCT clinics introduced the package to potential participants and invited the MSM to enroll as an IC after being diagnosed within one month. ICs were introduced to a choice of four modes for partner tracing and HIV testing, and they committed to

their preferred mode at enrollment. The VCT staff followed up with the ICs or their sexual partners until their sexual partners attended HTS during the three-month follow-up period.

Enrolled sexual partners proceeded to the next round if tested HIV positive and consented to be enrolled as ICs. The process ended when no positive HIV diagnosis was identified among the sexual partners, when a sexual partner was unreachable, or when no positive sexual partner consented to be enrolled as an IC.

The partner tracing and HIV testing package includes: 1) Couples' HIV counseling and testing (CHCT): The IC is requested to return to the VCT clinic with his sexual partner at a subsequent date to receive joint HIV testing and counseling. The IC is not requested to disclose his HIV status to his sexual partner prior to the CHCT session.; 2) Information assisted partner notification (IAPN): ICs provide the social media or phone contact information of their sexual partners. The VCT staff then contact the sexual partner through these communication tools, declare their own identity, name and working facility, by official landline/telephone or social software accounts, address sexual partner's risk of infection and encourage he/she to undertake HIV testing, communicate his/her risk of HIV infection; promote VCT; provide information of local VCT clinics; and facilitates a referral to a local VCT clinic; 3) Assisted HIV self-testing (HIVST): The VCT staff trains the ICs how to use an oral rapid HIV testing kit and provides them with testing kits; the ICs then train and provide their sexual partners with the oral rapid HIV testing kits, and return the testing kits to the clinic; 4) Patient referral: ICs are solely responsible for notifying sexual partners and encouraging them to receive HIV testing, and sexual partners come to VCT clinics by himself/herself to take HIV testing. In these modes, the VCT staff follow up to the IC or sexual partner once monthly until sexual partner attends HIV testing during the defined follow-up period of three months. S1 Fig shows this in more detail.

The partner tracing and HIV testing pilot program was implemented at all 23 Centers for Disease Control and Prevention (CDC) VCT clinic sites in the cities of Hangzhou and Ningbo from June 2014 to June 2016. The VCT staff received structured training (duration of two days) by the Zhejiang Provincial CDC. The training included information on the partner tracing and HIV testing package procedure, including how to elicit ICs and their sexual partners, and privacy concerns. All free HIV testing procedures and follow-up care followed the National Guideline for Detection of HIV/AIDS of China [25]. Participants who were tested negative will also be exposed to health education and behavior intervention during post-testing counselling. We utilized the RECORD guidelines to report this study [26].

**Data sources and variables.** Data on ICs and their sexual partners were collected by the VCT staff who received training on data collection and research ethics. VCT staff administered a short questionnaire on sexual partners and risk behaviors with ICs at VCT sites in addition to routine counseling after an HIV diagnosis.

The questionnaire included questions related to the IC demographics, sexual partner age and gender; the current status of their sexual relationship (currently in a relationship, not currently in a relationship, unsure); types of sexual relationships; frequency of sexual contact in the previous six months or even for stable and unstable relationships, respectively; and condom use in the previous six months. Stable relationships were defined by a report of a stable, non-commercial partner, or by having a male or female spouse. Unstable relationships were defined as either a reported commercial male partner, a casual, non-commercial male partner, or an unmarried female partner. We also collected data on the modes of the partner tracing and testing package selected by ICs (CHCT; Information assisted partner notification (IAPN); Assisted HIV-self testing (HIVST); Patient referral).

Data on sexual partners' HIV testing uptake and testing outcome were compiled from laboratory records.

The research team extracted data from a database provided by the VCT staff every three months.

## Statistics

We utilized EpiData 3.0 [Epidata.dk] for data entry and SAS 9.2 [SAS Institute Inc.] to conduct quantitative analyses. We presented ICs and sexual partners' characteristics and compared mode participation using frequencies and proportions for categorical variables. We calculated the mode preference of ICs by assigning a preference value of one to each IC and proportionally assigning this value across his selection for each sexual partner.

We compared HIV testing uptake (Tested sexual partners/Reachable sexual partners) and yield (HIV-positive sexual partners/Tested sexual partners) of sexual partners across different modes and presented the estimated effect sizes using odds ratios from generalized linear mixed models (GLMMs). In this model, we included data on sexual partner clustering within the same IC, sexual partner age, and sexual partner gender to control for potential correlation. P-values less than 0.05 were considered statistically significant.

## Ethics approval and consent to participate

This study was reviewed and approved by the Zhejiang Provincial Center for Disease Control and Prevention Ethics Review board (2013–001, 2018–038). Written informed consent forms were obtained from all participants in this study.

## Results

From June 2014 through June 2016, 2,495 newly diagnosed HIV-positive MSM from VCT clinics in Hangzhou and Ningbo were invited to participate in the study. 435 consented and enrolled as ICs, resulting in an enrollment of 17%. Following the program guidelines, the pilot resulted in two rounds of contact tracing and 4,176 disclosed sexual partners. Of those disclosed, 548 (13%) were reachable. The first-round ICs disclosed 4,116 sexual partners in total; 537 were reachable. Of those reached, 467 accepted HIV testing, 65 tested positive, and 11 consented to enroll as second-round ICs. These second-round ICs disclosed 60 sexual partners in total; 11 of these were reachable and accepted HIV testing. All reachable second-round partners accepted HIV testing, which resulted in nine HIV-positive test results. S2 Fig shows this in more detail.

There were 367 ICs who reached 1 sexual partner, 61 ICs who reached 2 sexual partners, 14 ICs who reached 3 sexual partners, 3 ICs who reached 4 sexual partners, and 1 IC who reached 5 sexual partners. Overall, the proportion of HIV testing among reachable sexual partner for HIV testing was 87%, and HIV testing yield among tested sexual partners was 16%.

In total, 446 first- and second-round ICs enrolled in the pilot, with a mean age of 31.5 (SD = 9.6) years old. Around half of ICs had a college education or above (212, 48%), lived in the local county of the VCT (225, 52%), and disclosed six or more sexual partners (239, 54%). The two rounds of tracing and testing in the pilot resulted in 4,176 disclosed sexual partners (1,014 in recent 6 months), and 548 reached sexual partners (Fig 1). The mean age of sexual partners was 30.7 (SD = 8.9) years old, and most were male (460, 84%). Nearly half of the reachable sexual partners (267, 49%) were currently in a sexual relationship with ICs. Among reachable sexual partners in a stable relationship with an IC, over half (179, 60%) had sex with ICs less than once per week. Among those in casual relationships, almost half (110, 45%) had sex with ICs a total of two to five times. Only approximately one-third of reachable sexual partners (34.7%) reporting consistently using condoms with ICs over the past six months (Table 1).

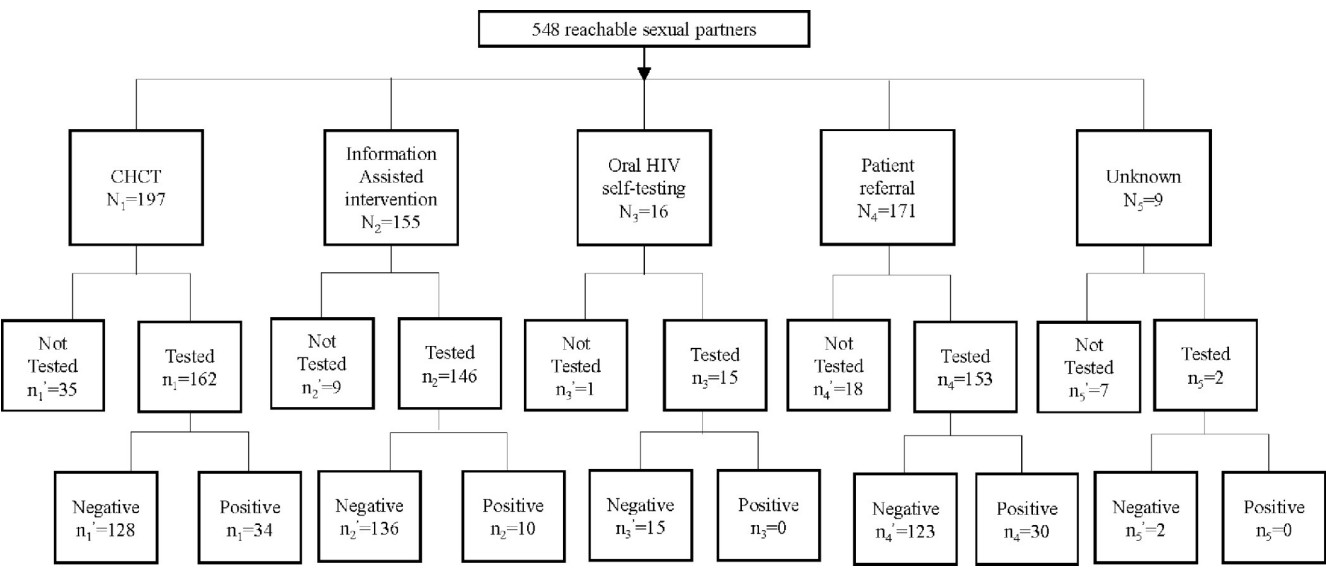

**Fig 1. Cascade of partner tracing and HIV testing package among sexual partners of newly diagnosed HIV positive men who have sex with men June 2014 through June 2016 in Hangzhou and Ningbo, China.**

ICs selected the patient referral and CHCT mode most frequently (163, 37%) and the HIVST mode the least (13, 3%) (Table 2).

HIV testing uptake among all reachable sexual partners was 87% and varied between 82–94% across the selected modes. Sexual partners reached through the IAPN mode showed the highest testing uptake (94%).

HIV testing yield among all tested sexual partners was 16%. Among the selected testing and tracing modes, patient referral yielded 18%, CHCT 17%, and IDAPN 7% (Table 2).

We found the odds of a reachable sexual partner taking an HIV test were 290% greater for contacts who were traced and tested through IAPN compared to those traced through CHCT (95% CI: 1.6, 9.3). The odds of a reachable sexual partner testing HIV-positive were 90% lower for contacts who were traced and tested in IAPN compared to those traced through CHCT (95% CI: 0.0, 0.6) (Table 3).

Regarding different types of sexual relationship with sexual partners, 43% (93/219) of stable same-sex contacts and 55.1% (43/78) opposite-sex spouses were designated to CHCT; 40% (95/236) of casual same-sex partners were designated to IAPN. In casual, non-commercial same-sex relationships, individuals traced and tested through IAPN who received HIV testing were present in higher proportions than individuals traced and tested through CHCT who received HIV testing. S1 Table shows this in more detail.

## Discussion

We presented results from a pilot program on combination partner tracing and HIV testing to expand HIV testing uptake for sexual partners of HIV-positive MSM. Our results showed that this package is a feasible approach in this setting, which may help target and locate high-risk populations.

We found that a high proportion (87%) of reachable, sexual partners were successfully tested, which was higher than a previous study in eastern China (23%), but similar to a study in Hangzhou and Kunming (85%) in the Yunnan province southwest of China [27,28]. This suggests that a potential advantage of this partner tracing and testing package may be the

**Table 1. Newly diagnosed HIV-positive men who have sex with men enrolled in a partner tracing and HIV testing program, from June 2014 through June 2016 in the cities of Hangzhou and Ningbo, China.**

| Characteristics | N | % |
|---|---|---|
| Index cases of newly diagnosed men who have sex with men (MSM) | 446 | 100.0 |
| Index cases recruited through self-testing at VCT sites | 435 | 97.5 |
| Index cases recruited through partner tracing and HIV testing program | 11 | 2.5 |
| Age of index case, years | | |
| 18–24 | 108 | 24.2 |
| 25–34 | 200 | 44.8 |
| 35–44 | 78 | 17.5 |
| 45 and above | 48 | 10.8 |
| Missing | 12 | 2.7 |
| Education level completed by index case | | |
| Primary school or below | 31 | 7.0 |
| Junior middle school | 87 | 19.5 |
| Senior high school | 104 | 23.3 |
| College or above | 212 | 47.5 |
| Missing | 12 | 2.7 |
| Residence of index case | | |
| local county | 225 | 51.8 |
| outside local county | 209 | 48.2 |
| Number of sexual partners disclosed by index case | | |
| 1 | 39 | 8.7 |
| 2–5 | 168 | 37.7 |
| 6–10 | 119 | 26.7 |
| 11 and above | 120 | 26.9 |
| Number of sexual partners disclosed by index case in recent 6 months | | |
| 0–1 | 206 | 46.2 |
| 2–5 | 212 | 47.5 |
| 6–10 | 21 | 4.7 |
| 11 and above | 7 | 1.6 |
| Mode preference of index case, weighted[a] | | |
| CHCT | 160 | 35.9 |
| IAPN | 126 | 28.3 |
| HIVST | 13 | 2.9 |
| Patient referral | 139 | 31.2 |

MSM: men who have sex with men; CHCT: couples' HIV testing and counseling; IAPN: information assisted partner notification; HIVST: assisted HIV self-testing

[a] Weighted by assigning a preference value of one to each IC and proportionally assigning this value across his selection for each sexual partner

ability to expand HIV testing uptake among reachable sexual partners of HIV-positive MSM, and highlight that further research on the effectiveness of testing uptake is needed.

We found a higher HIV testing yield (16%) among tested sexual partners of HIV-positive MSM compared to the HIV prevalence of the Zhejiang province general population (estimated at 0.04%) [13] and general population passive tracing (VCT) testing yield of its cities (1.30% in Hangzhou) [29]. These results were not unexpected, given the higher risk behaviors of MSM [7,30,31]. Our testing yield result was also higher than that of MSM in a city in Zhejiang province who were contacted through passive tracing (VCT; 4.19% in Lishui, 2008–2015) [32]. A

**Table 2. Sexual partners of newly diagnosed HIV-positive men who have sex with men enrolled in a partner tracing and HIV testing program from June 2014 through June 2016, in the cities of Hangzhou and Ningbo, China.**

| Characteristics | N | % |
|---|---|---|
| Reachable sexual partners reported by index cases | 548 | |
| No enrollment as index case in partner tracing and HIV testing program | 537 | 98.0 |
| Subsequent enrollment as index case in partner tracing and HIV testing program | 11 | 2.0 |
| Age of sexual partners, years | | |
| 17–24 | 125 | 22.8 |
| 25–34 | 272 | 49.6 |
| 35–44 | 105 | 19.2 |
| 45 and above | 46 | 8.4 |
| Gender of sexual partners | | |
| Male | 460 | 83.9 |
| Female | 85 | 15.5 |
| Missing | 3 | 0.6 |
| Current relationship status with sexual partners, as disclosed by index case* | | |
| Currently in a relationship | 267 | 48.7 |
| Not sure | 11 | 2.0 |
| No current relationship | 267 | 48.7 |
| Missing | 3 | 0.6 |
| Type(s) of sexual relationship with sexual partners, as disclosed by index case cases | | |
| *Stable relationship with sexual partner* | | |
| Stable, non-commercial same-sex relationship | 219 | 40.0 |
| opposite-sex spouse | 78 | 14.2 |
| *Non-Stable relationship with sexual partner* | | |
| Commercial same-sex relationship | 1 | 0.2 |
| Casual, non-commercial same-sex relationship | 236 | 43.1 |
| Unmarried opposite-sex relationship | 7 | 1.3 |
| Other | 7 | 1.3 |
| Frequency of sex as disclosed by index cases | | |
| *Stable relationship with sexual partner*: times per week in the past 6 months | 297 | |
| <1 | 176 | 59.3 |
| 1–2 | 95 | 32.0 |
| 3–4 | 12 | 4.0 |
| 5 and above | 0 | 0 |
| *Non-Stable relationship with sexual partner*: total sexual partner | 244 | |
| 1 | 76 | 31.1 |
| 2 | 70 | 28.7 |
| 3–4 | 40 | 16.4 |
| 5 and above | 33 | 13.5 |
| Condom use between index case and sexual partner in the past 6 months, as disclosed by index case | | |
| Never | 91 | 16.6 |
| Inconsistent | 256 | 46.7 |
| Consistent | 190 | 34.7 |
| Missing | 11 | 2.0 |

previous study also found that partner tracing and testing may be more effective than other HIV testing policies in identifying undiagnosed infections [28]. These findings suggest that a

**Table 3. HIV testing uptake and yield by modes of partner tracing and testing package among reachable sexual partners of newly diagnosed HIV positive MSM in June 2014 through June 2016, in the cities of Hangzhou and Ningbo, China.**

| Characteristics | | | Tested | | Not tested | | OR | (95% | CI) | Neg | | Pos | | OR | (95% | CI) |
|---|---|---|---|---|---|---|---|---|---|---|---|---|---|---|---|---|
| | N | (%) | n | (%) | n | (%) | | | | n | (%) | n | (%) | | | |
| Reachable sexual partners | 548 | ((100.0) | 478 | ((87.2) | 70 | ((12.8) | | | | 404 | (85.6) | 74 | ((15.5) | | | |
| *Modes* | | | | | | | | | | | | | | | | |
| CHCT | 197 | ((35.9) | 162 | ((82.2) | 35 | ((17.8) | 1.0 | | | 128 | ((79.0) | 34 | ((21.0) | 1.0 | | |
| IAPN | 155 | ((28.3) | 146 | ((94.2) | 9 | ((5.8) | 3.9 | (1.6, | 9.3) | 136 | ((93.2) | 10 | ((6.8) | 0.1 | (0.0, | 0.6) |
| HIVST | 16 | ((2.9) | 15 | ((93.8) | 1 | ((6.3) | 3.4 | (0.4, | 30.0) | 15 | ((100.0) | 0 | ((0.0) | - | | |
| Patient Referral | 171 | ((31.2) | 153 | ((89.5) | 18 | ((10.5) | 1.8 | (0.9, | 3.6) | 123 | ((80.4) | 30 | ((19.6) | 0.9 | (0.4, | 2.1) |
| Unknown | 9 | ((1.6) | 2 | ((22.2) | 7 | ((77.8) | - | | | 2 | ((100.0) | 0 | ((0.0) | - | | |

OR: Odds Ratio; CI: Confidence Interval; Neg: negative HIV test; Pos: positive HIV test; CHCT: couples' HIV testing and counseling; IAPN: Information assisted partner notification; HIVST: assisted HIV self-testing

higher efficiency of integrating active case-finding strategies with passive strategies is necessary to target MSM contacts in this setting.

Our results showed variability in the testing uptake and yield across different tracing and testing modes. In the CHCT mode, reachable sexual partners presented the lowest rate of HIV testing uptake, and less odds of uptake than those in the IAPN mode. As CHCT requires that couples take HIV testing and counseling together [18], fear of disclosure or discrimination may be barriers to selection of this mode [33]. Our results on CHCT uptake (82%) are similar to those of a previous study in China that showed high willingness to receive CHCT (86%) [34]. However, this mode resulted in effective case finding compared to other modes, which indicates that it may be more acceptable for same-sex couples who are in or seeking committed relationships [35]. Our results also showed that CHCT is more acceptable among stable sexual partners. These individuals are more likely to engage in unprotected anal sex, and are thus at higher risk of HIV transmission [36], which may contribute to our high-yield results for those who received testing using this mode. Therefore, CHCT may be feasible to identify undiagnosed infections in stable sexual partners.

The information assisted partner notification presented the highest HIV testing uptake. It is in-line with previous studies that showed acceptability of similar methods [15]. Because this mode did not require direct contact between ICs and sexual partners, it may be effective in targeting sexual partners with limited contact information (such as those with social media contact information only) and protect the privacy of both sides. Information assisted partner notification may result in high testing uptake when ICs do not have the intention or ability to notify sexual partners [21]. However, in our study, case finding in the information assisted partner notification was not as effective as that for CHCT and patient referral. This may reflect the fact that the choice of information assisted mode may indicate a less intimate relationship between ICs and their sexual partners, resulting in the difficulty for ICs to reach their sexual partners.

Previous studies showed that oral HIV self-testing can be feasible and effective for MSM and their sexual partners [37]. However, our results indicated that fewer participants chose this mode for their sexual partners. As ICs needed to learn how to use oral HIV self-testing kits and teach this skill to their sexual partners, the overly complex instructional materials and procedure may have restricted ICs' intention to bring kits to their partners [38]. Further studies and practical applications are needed to identify whether oral HIV self-testing is feasible for expanding HIV testing in the sexual partners of HIV-positive MSM in this setting. Results

showed that patient referral is still an important component in expanding HIV testing and case-identification among MSM and should continue to be an option among multi-modal HIV testing uptake programs.

There are some limitations to this research. We utilized self-reported data from ICs, which could have caused information bias in the characteristics of sexual behavior between ICs and sexual partners. In addition, enrollment in the study was limited, as only 435 of 2508 total newly diagnosed HIV positive MSM agreed to participate, and only 18% (548/4,116) of disclosed sexual partners were traceable. The rate of participation as index cases varied among different types of HIV positive populations, and the rate of participation in our study might be lower, compared to results of other studies [39, 40]. However, the number of sexual partners in recent six months (1,014) was much less than the number of sexual partners in their whole life (4,116), which is also need to be considered, as we assume that those reachable sexual partners may mostly come from those who are still in touch with ICs in recent six months. We have compared those HIV positive MSM who enrolled to those who did not, and have found that those who settled in the detection place vs. those outside (28.5% vs. 12.3%), those with education level of college or above vs. those with lower education level (20.5% vs. 16.0%) would be more likely to be enrolled to the program, and there was no statistical difference for the variables of age and marital status. The demographic characteristics of HIV positive MSM might be associate with their participation in this pilot study. Previous studies showed that fear of disclosure of infection status, lack of contact information of sexual partners, and other limitations still play important roles in the gap between numbers of total newly diagnosed HIV positive MSM and enrolled as ICs, and the gap between numbers of self-disclosed sexual partners and reachable sexual partners [28, 41]. However, this pilot package is feasible for identifying undiagnosed infections among reachable sexual partners of HIV-positive MSM. Therefore, our findings are still significant as they provide insight into future approaches.

## Conclusions

In conclusion, a partner tracing and testing package program that includes CHCT and information assisted partner notifications may have the potential to play an important role in expanding HIV testing uptake among sexual partners of HIV-positive MSM. This package is also a feasible approach for case finding in high-risk populations. An information assisted partner notification may be an acceptable option to reach sexual partners for whom limited contact information is available. However, further study on the feasibility of oral HIV self-testing among sexual partners of MSM is needed.

## Supporting information

**S1 Fig. Introduction for modes in the partner tracing and HIV testing package among sexual partners of newly diagnosed HIV positive men who have sex with men, Zhejiang Province, China.**
(PDF)

**S2 Fig. Two round procedure of partner tracing and HIV testing package among sexual partners of newly diagnosed HIV positive men who have sex with men June 2014 through June 2016 in Hangzhou and Ningbo, China.**
(PDF)

**S1 Table. HIV testing uptake by modes in different types of sexual relationship among reachable sexual partners of newly diagnosed HIV positive MSM in June 2014 through**

**June 2016, in Hangzhou and Ningbo Cities, China.**
(PDF)

**S1 File.**
(DOCX)

**S2 File.**
(DOCX)

## Acknowledgments

This research was conducted through the Structured Operational Research and Training Initiative (SORT-IT), a global partnership coordinated by the Special Programme for Research and Training in Tropical Diseases at the World Health Organization (WHO/TDR). The training model is based on a course developed jointly by the International Union Against Tuberculosis and Lung Disease (The Union) and Médecins Sans Frontières (MSF). The Specific SORT-IT programme which resulted in this publication was implemented by: Médecins Sans Frontières, Brussels Operational Centre, Luxembourg and the China Centre for Disease Control & Prevention. Mentorship and the coordination/facilitation of this SORT-IT workshop were provided through the University of Washington, Department of Global Health, USA; AMPATH, Eldoret, Kenya; Sustainable Health Systems, Sierra Leone; Universidad Pontificia Bolivariana, Columbia; Global AIDS Interfaith Alliance, USA; Centre for Operational Research, The Union, Paris, France; and the China Centre for Disease Control & Prevention. Thanks to all CDC staff in Hangzhou and Ningbo cities.

## Author Contributions

**Conceptualization:** Xiaohong Pan, Qiaoqin Ma, Shichang Xia.

**Formal analysis:** Mingyu Luo, Katrina Hann, Guomin Zhang.

**Investigation:** Jun Jiang, Lin Chen.

**Methodology:** Jun Jiang, Lin Chen.

**Project administration:** Xiaohong Pan.

**Software:** Mingyu Luo.

**Supervision:** Xiaohong Pan, Qiaoqin Ma, Shichang Xia.

**Validation:** Xiaohong Pan.

**Writing – original draft:** Mingyu Luo, Katrina Hann, Guomin Zhang.

**Writing – review & editing:** Mingyu Luo, Katrina Hann, Guomin Zhang.

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
