## [Decision Letter · Decision Letter 0]

12 Dec 2019

PONE-D-19-30003

HIV testing uptake and yield among sexual contacts of HIV-positive men who have sex with men in Zhejiang Province, China, 2014-2016: a cross-sectional pilot study of a choice-based partner tracing and testing package

PLOS ONE

Dear Mrs Pan,

Thank you for submitting your manuscript to PLOS ONE. After careful consideration, we feel that it has merit but does not fully meet PLOS ONE’s publication criteria as it currently stands. Therefore, we invite you to submit a revised version of the manuscript that addresses the points raised during the review process.

Please address all three reviewers' comments point-by-point. In particular, all reviewers have concerns with English writing. Please have a native English speaker copy-edit it or use copying-editing services. PLoS One will not be able to accept manuscripts that are written in non-standard English. Secondly, both reviewers 2 and 3 raised concerns with the analytical approach and I agree that controlling for correlations is needed. 

We would appreciate receiving your revised manuscript by Jan 26 2020 11:59PM. To enhance the reproducibility of your results, we recommend that if applicable you deposit your laboratory protocols in protocols.io, where a protocol can be assigned its own identifier (DOI) such that it can be cited independently in the future. For instructions see: http://journals.plos.org/plosone/s/submission-guidelines#loc-laboratory-protocols

We look forward to receiving your revised manuscript.

Kind regards,

Chongyi Wei, DrPH

Academic Editor

PLOS ONE

Journal Requirements:

2. Please include additional information regarding the survey or questionnaire used in the study and ensure that you have provided sufficient details that others could replicate the analyses. For instance, if you developed a questionnaire as part of this study and it is not under a copyright more restrictive than CC-BY, please include a copy, in both the original language and English, as Supporting Information. In addition, please ensure you include any details of pre-testing of this questionnaire, i.e. how many participants were involved and from where were they recruited.

3. Please state what type of consent was obtained from guardians of minors included in this study, ensure that you have specified (1) whether consent was informed and (2) what type you obtained (for instance, written or verbal, and if verbal, how it was documented and witnessed).

4. 

Thank you for stating the following in the Acknowledgments Section of your manuscript:

"Funding for the costs of publication in an open-access, peer-reviewed journal was supported by Key Project on Social Development among S&T Major Project of Zhejiang Province, China (2013C03047-1), Zhejiang Provincial Medicine Science and Technology Plan(2015PYA004), National Science and Technology Major Project of China (2017ZX10201101), The Training Project of Young Scientific and Technological Innovative Talents of Zhejiang Provincial Center of Disease Control and Prevention."

6. Your ethics statement must appear in the Methods section of your manuscript. If your ethics statement is written in any section besides the Methods, please move it to the Methods section and delete it from any other section. Please also ensure that your ethics statement is included in your manuscript, as the ethics section of your online submission will not be published alongside your manuscript.

Reviewers' comments:

Reviewer's Responses to Questions

**Comments to the Author**

1. Is the manuscript technically sound, and do the data support the conclusions?

Reviewer #1: Yes

Reviewer #2: Partly

Reviewer #3: Yes

2. Has the statistical analysis been performed appropriately and rigorously? 

Reviewer #1: Yes

Reviewer #2: No

Reviewer #3: Yes

3. Have the authors made all data underlying the findings in their manuscript fully available?

Reviewer #1: Yes

Reviewer #2: No

Reviewer #3: No

4. Is the manuscript presented in an intelligible fashion and written in standard English?

Reviewer #1: No

Reviewer #2: Yes

Reviewer #3: No

5. Review Comments to the Author

Reviewer #1: Congratulations on a fascinating project. This paper articulates the study design and the relevant findings. Also, the methods are sound. You could however explain a little more fully and in detail the different methods of partner tracing (i.e. not all readers will understand patient referral without an explanation).

My greatest concern with this paper is the written English. The paper needs to be revised by a native English speaker/writer which will allow these important findings to be understood more clearly and easily by the relevant audience.

Reviewer #2: This study used routinely collected data to evaluate a contact tracing and testing intervention for partners of men who have sex with men in Zhejiang province, China conducted over two years. Overall, the manuscript read well, but I have concerns about the analysis, intervention description, and the interpretation.

1. The authors conclude that the partner tracing and testing package was “effective for case-finding among a high-risk population.” However, it is unclear what criteria were used to determine “effectiveness”, as there were no outcomes measures for a comparison group that did not receive the intervention. Moreover, the authors’ optimistic appraisal does not seem very strongly supported by the data. The arguably low participation in the intervention (18%) and arguably low proportion of reachable contacts (12%), would lead many readers to conclude that the intervention was in fact not very effective. What if the participation rate and proportion of reachable contacts were only 5%? What was the cut-off point defining effectiveness, if any? Would the authors still conclude that the intervention was “effective”, as long as a relatively high proportion of reachable contacts of completed testing? Please provide stronger justification for why the intervention should be considered “effective”, or modify the appraisal regarding the effectiveness of the intervention. In addition, further discussion is needed about why the participation and reachable contact rates were so low, and what can be done to improve greater uptake.

2. In the interest of implementation and replication, much more detail about the intervention (each mode) is needed. For example, for the “information-driven assisted partner notification” mode of intervention, how did staff introduce themselves when reaching out to contacts of the indexes? What was their training? How were they able to ensure confidentiality? Who covered the cost of testing of partners? For the “patient referral” intervention mode, how did study investigators ascertain the HIV test results of their contacts?

3. In the abstract and text, it appears that the authors did not meaningfully differentiate in their presentation of statistically significant and non-statistically significant results. For example, in the abstract: “odds of a reachable sexual contact enrolled in 28 information-driven mode taking an HIV test were 90% more than that of one enrolled in 29 patient referral (95%CI:0.8, 4.4).” Why conduct significance tests and state “We assigned statistical significance at p-values less than 0.05”, if the presentation of significant results are basically the same as non-significant results?

4. Given the fact that partner contacts were clustered within indexes, I am concerned that the analytic approach did not control for correlated data. Please reanalyze the data using an analytic approach that controls for clustering of observations within indexes (e.g., GEE or random effects), or provide strong justification for why such an analysis is unnecessary.

5. Numerous grammatical errors need to be corrected. For example, “We defined reachable sexual contacts as those can be contacted…” (pg 7).

Reviewer #3: This paper was well conceptualized, however, there are instances where the writing does not follow standard English grammatical conventions. It may be prudent of the authors to have a proof reader who can ensure that sentence structure and the usage of punctuation are sound.

6. PLOS authors have the option to publish the peer review history of their article (what does this mean?). If published, this will include your full peer review and any attached files.

Reviewer #1: No

Reviewer #2: No

Reviewer #3: No

---

## [Author Response · Author response to Decision Letter 0]

6 Mar 2020

Dear editor and reviewers,

We appreciate it for your efforts to review this manuscript.

Thank you for the reviewers’ comments. We have revised the manuscript and especially paid attention to your comments and suggestions.

We have changed “sexual contact” to “sexual partner” in this revised manuscript as we think this description is much more accurate to define a person who had sex with IC.

In this revised version. Mr. Qiaoqin Ma has helped a lot with improving the description and related analysis. We authors have agreed that he should be listed as co-corresponding author.

Answers to reviewers’ comments are as follows:

Reviewer #1: 

Congratulations on a fascinating project. This paper articulates the study design and the relevant findings. Also, the methods are sound. You could however explain a little more fully and in detail the different methods of partner tracing (i.e. not all readers will understand patient referral without an explanation).

My greatest concern with this paper is the written English. The paper needs to be revised by a native English speaker/writer which will allow these important findings to be understood more clearly and easily by the relevant audience.

Answer: Thanks for the reviewer’s comments. We have revised the method section, and provided details about different modes of partner tracing and HIV testing package. 

About the written English, we have asked a native English speaker to help revise the manuscript and edit the language. We also have searched for language editing service from www.editage.cn.

Reviewer #2:

1. The authors conclude that the partner tracing and testing package was “effective for case-finding among a high-risk population.” However, it is unclear what criteria were used to determine “effectiveness”, as there were no outcomes measures for a comparison group that did not receive the intervention. Moreover, the authors’ optimistic appraisal does not seem very strongly supported by the data. The arguably low participation in the intervention (18%) and arguably low proportion of reachable contacts (12%), would lead many readers to conclude that the intervention was in fact not very effective. What if the participation rate and proportion of reachable contacts were only 5%? What was the cut-off point defining effectiveness, if any? Would the authors still conclude that the intervention was “effective”, as long as a relatively high proportion of reachable contacts of completed testing? Please provide stronger justification for why the intervention should be considered “effective”, or modify the appraisal regarding the effectiveness of the intervention. In addition, further discussion is needed about why the participation and reachable contact rates were so low, and what can be done to improve greater uptake.

Answer: Thanks for reviewer’s comments. We have seriously taken this comment into consideration. One important reason for overall low uptake might be that the number of self-disclosed sexual partners was the number of total sexual partners during one IC’s life. We consider that those reachable sexual partners may mostly come from those who are still in touch with ICs in recent period of time, thus we analyze the number of sexual partners in recent 6 months disclosed by index case, and to reflect the real gap between disclosed sexual partners and reachable sexual partners in the limitation part of discussion section. The participation rate of index cases varied among different types of HIV positive populations.

In this revised manuscript, we have used “feasible” in place of “effective” (lines 343-366).

2. In the interest of implementation and replication, much more detail about the intervention (each mode) is needed. For example, for the “information-driven assisted partner notification” mode of intervention, how did staff introduce themselves when reaching out to contacts of the indexes? What was their training? How were they able to ensure confidentiality? Who covered the cost of testing of partners? For the “patient referral” intervention mode, how did study investigators ascertain the HIV test results of their contacts?

Answer: Many thanks. 

We have revised the method section. All the details about introduction process, confidentiality, training, cost of testing, ascertaining testing results have been provided in the method section (lines 146-165), and we have updated the supplemental figure 1 to show this in detail.

All our co-authors have agreed that the English translation for “information-driven assisted partner notification” mode is redundant, and have decided to use “information assisted partner notification” to describe this mode.

3. In the abstract and text, it appears that the authors did not meaningfully differentiate in their presentation of statistically significant and non-statistically significant results. For example, in the abstract: “odds of a reachable sexual contact enrolled in 28 information-driven mode taking an HIV test were 90% more than that of one enrolled in 29 patient referral (95%CI:0.8, 4.4).” Why conduct significance tests and state “We assigned statistical significance at p-values less than 0.05”, if the presentation of significant results are basically the same as non-significant results?

Answer: We have revised related section, changed reference group in this categorical variable (modes), and changed the description, in related sections (lines 28-30, 265-269) based on reviewer’s comments.

4. Given the fact that partner contacts were clustered within indexes, I am concerned that the analytic approach did not control for correlated data. Please reanalyze the data using an analytic approach that controls for clustering of observations within indexes (e.g., GEE or random effects), or provide strong justification for why such an analysis is unnecessary.

Answer: Thanks, we have seriously considered this comment. 

Now, we used univariate logistic regression to compare HIV testing uptake and yield of sexual contacts across different modes. In this study, we found that 18% (79/446) ICs have more than 1 reachable sexual partners, therefore we agree that potential correlation within clustered sexual partners should be considered in analysis process.

The estimated effect size now is produced and represented using odds ratios through generalized linear mixed models (GLMMs). In this model, whether or not sexual partners were clustered with the same IC, sexual partners’ age, and sexual partners’ gender are included to control potential correlation. We have revised related sections (lines 28-30, 205-211).

5. Numerous grammatical errors need to be corrected. For example, “We defined reachable sexual contacts as those can be contacted…” (pg 7).

Answer: Thanks, we have revised the whole manuscript to correct these grammatical errors, and we have searched for language editing service from www.editage.cn to improve the description.

Reviewer #3:

This paper was well conceptualized, however, there are instances where the writing does not follow standard English grammatical conventions. It may be prudent of the authors to have a proof reader who can ensure that sentence structure and the usage of punctuation are sound.

Answer: Thanks for the reviewer’s comments. We have revised the written English through the whole manuscript. We also ask a native English speaker to help revise the manuscript, and we have searched for language editing service from www.editage.cn to improve the description.

Methods

• Please explicitly state the time frame (e.g., past month, past 2 months, etc.) in which someone would be considered “newly diagnosed as HIV positive”.

Answer: Thanks for the reviewers’ comments. Now we explicitly state the time frame as being diagnosed in one month in which someone would be considered “newly diagnosed as HIV positive”. We have revised the method section to make the description clearer, in lines 108, 137-138.

• The sentence in lines 109-111 is a bit hard to understand.

Answer: This sentence was to state the definition of “reachable sexual contacts”. We have revised this sentence to make the expression clearer. The revised version is “We defined reachable sexual contacts as those who could be contacted by face-to-face, official landline/telephone, social software (such as QQ, WeChat and Blued) or other communication tools.” In lines 117-119 now.

• Were sexual contacts of IC who were under the age of 18 and who were HIV-negative also included in this study? You explicitly mentioned sexual contacts under 18 who were HIV-positive.

Answer: Yes, there was one male sexual partner of IC, who was 17 years old, and HIV negative, was included in this study. 

We provide standard post-testing counselling, and all participants tested negative would be exposed to health education and behavior intervention. We have made it clear in lines 173-175.

• Given the nature of this study, it may be important to include the IC’s viral load data in this analysis, since this metric is often used to estimate potential risk of forward transmission.

Answer: In China, ARV and viral load testing is provided to HIV positive patients free of charge by the government. We couldn’t collect viral load data of all newly diagnosed IC because viral load testing for free is not required to be done at the moment when a patient is newly diagnosed, just required to be provided once a year to the patients.

• What type of regression did you use? You presented odds ratios, which makes me assume logistic regressions were used.

Answer: Yes, we used univariate logistic regression to compare HIV testing uptake and yield of sexual partners across different modes.

In this revised manuscript, we have updated the analysis through generalized linear mixed models (GLMMs) to control potential correlation within clustered sexual partners (lines 205-211).

• It may be worthwhile to examine if sexual contacts who are in different types of relationships prefer certain methods of contact. For example, if the sexual contact is in a relationship with someone of the opposite sex, the sexual contact may be hesitant to appear with his male sexual partner in public.

Answer: Yes, we agree that sexual partners who are in different types of relationships prefer certain methods of contact. In this study, four modes for partner tracing and HIV testing were introduced to each IC, who then choose a mode which who think that this is the best and appropriate to him and his sexual partners.

On the other hand, unlike the western, two Chinese men may not hesitate to appear together in public because it won’t bring any suspicions that they are MSM or sex partners in Chinese culture as long as they don't act intimately even though one of them is in a relationship with someone of the opposite sex.

• Were there processes in place to make sure that sexual contacts were not duplicated? For example, if two different ICs identified the same sexual contact, this may affect the validity of the post estimates.

Answer: Thanks for your comment. 

This study was conducted at 23 VCT clinics of 23 local CDCs in Hangzhou and Ningbo, 2 big cities in Zhejiang Province, China. Standardized VCT service is provided to each VCT seeker at VCT clinics, which is professional, confidential, free of charge and client friendly. Theoretically, one partner in this study is less likely to go to another VCT clinics again after he/she visited one clinic requested by the study staff. Further, we didn’t find duplicated sexual partners who is HIV negative using IC’s partners information such as social software accounts, phone numbers to match each other. 

 All HIV positive person must be reported to China national HIV/AIDS reporting system. Verification of duplicate case reporting is done when one case is reported. Thus, duplicate IC or IC’s HIV positive partner in this study could not be an issue to be noticed.

• It may be important to explain exactly why those with severe AIDS complications were excluded. If the authors considered high viral load/low cd4 count as a severe complication, those at highest risk for forward transmission may be excluded from this study.

Answer: In inclusion and exclusion section, we have revised this section into “severe AIDS complication or other illnesses requiring hospitalization”, in lines 112-113, to make it much clearer.

Results

• The first three 95% Cis presented in lines 222-226 all cross the null, however, they are presented as if they were statistically significant.

Answer: Based on the reviewers’ comment, we changed reference group in this categorical variable (modes), reanalyzed the dada and have made revision in related sections (lines 275-281).

Discussion

• You mention that only a small proportion of potential participants enrolled in this study. Would it be possible to compare those who enrolled to those who did not to identify if there are certain characteristics that differ between the two groups?

Answer: Thanks for your comment. 

We couldn’t collect the characteristics about IC’s sexual partners who are not reachable. Therefore, we were not able to compare the differences between those IC who enrolled and those who did not.

We could just collect social-demographics for IC, such as age, education status, marital status and residence. We have compared those HIV positive MSM who enrolled to those who did not, and have found that those who settled in the detection place vs. those outside ( 28.5% vs. 12.3%) , those with education level of college or above vs. those with lower education level ( 20.5% vs. 16.0%) would be more likely to be enrolled to the program, and there was no statistical difference and there was no statistical difference for the variables of age and marital status. This comparison has also been stated in the discussion section as a limitation. (lines 353-358)

Minor

Line 186 – Change 4176 to 4,176

Line 187 – please add a space between 548 and (13%)

Line 195 – please add a space between 31.5 and (SD=9.6)

Line 198 – please add a space between 30.7 and (SD=8.9)

It may be worthwhile for the authors to go through the manuscript to identify additional grammatical errors.

Answer: Thanks for your advices, we have revised related sections and go through the manuscript to revise other grammatical errors.

Additional requirements

Answer: Thank you for your advice. We have downloaded these files and checked the format.

2. Please include additional information regarding the survey or questionnaire used in the study and ensure that you have provided sufficient details that others could replicate the analyses. For instance, if you developed a questionnaire as part of this study and it is not under a copyright more restrictive than CC-BY, please include a copy, in both the original language and English, as Supporting Information. In addition, please ensure you include any details of pre-testing of this questionnaire, i.e. how many participants were involved and from where were they recruited.

Answer: We have uploaded two copies of the questionnaire, one in Chinese and the other one in English, as supporting files.

Pre-investigation was conducted among 20 HIV positive MSM in Hangzhou city to improve the quality of questionnaire, mainly in questions related to sexual contacts with their sexual partners.

3. Please state what type of consent was obtained from guardians of minors included in this study, ensure that you have specified (1) whether consent was informed and (2) what type you obtained (for instance, written or verbal, and if verbal, how it was documented and witnessed).

Answer: Consent was informed and written informed consent forms were obtained from HIV positive minor participants and their guardians (lines 121-123). We make it clear in the method section.

"Funding for the costs of publication in an open-access, peer-reviewed journal was supported by Key Project on Social Development among S&T Major Project of Zhejiang Province, China (2013C03047-1), Zhejiang Provincial Medicine Science and Technology Plan(2015PYA004), National Science and Technology Major Project of China (2017ZX10201101), The Training Project of Young Scientific and Technological Innovative Talents of Zhejiang Provincial Center of Disease Control and Prevention."

Answer: Thanks. We have removed details of funding information from the manuscript to the online submission form.

The funding statement will be updated as follows:

“This study was supported by Key Project on Social Development among S&T Major Project of Zhejiang Province, China (2013C03047-1), Zhejiang Provincial Medicine Science and Technology Plan (2015PYA004), National Science and Technology Major Project of China (2017ZX10201101), The Training Project of Young Scientific and Technological Innovative Talents of Zhejiang Provincial Center of Disease Control and Prevention.

Answer: We are sorry we couldn’t upload the database in the system because this database contains personal and sexual information of HIV positive persons, which is sensitive, confidential and under restrict management by Chinese Law. Please contact our ethics committee if having any question about this.

Contact information of Zhejiang Provincial Center for Disease Control and Prevention Ethics Review board: 

Address: NO.3399, Binsheng Road, Binjiang District, Hangzhou City, Zhejiang Province, China. 

Please contact: Mr. Zhenggang Jiang

Email: zhgjiang@cdc.zj.cn，TEL：+86 571-87115105

6. Your ethics statement must appear in the Methods section of your manuscript. If your ethics statement is written in any section besides the Methods, please move it to the Methods section and delete it from any other section. Please also ensure that your ethics statement is included in your manuscript, as the ethics section of your online submission will not be published alongside your manuscript.

Answer: We have checked and moved ethics statement to the methods section, and deleted ethics statement in any other sections.

---

## [Decision Letter · Decision Letter 1]

16 Mar 2020

PONE-D-19-30003R1

HIV testing uptake and yield among sexual partners of HIV-positive men who have sex with men in Zhejiang Province, China, 2014-2016: a cross-sectional pilot study of a choice-based partner tracing and testing package

PLOS ONE

Dear Mrs Pan,

Thank you for submitting your manuscript to PLOS ONE. After careful consideration, we feel that it has merit but does not fully meet PLOS ONE’s publication criteria as it currently stands. Therefore, we invite you to submit a revised version of the manuscript that addresses the points raised during the review process.

Please address Reviewer# 2's additional comments.

We would appreciate receiving your revised manuscript by Apr 30 2020 11:59PM. To enhance the reproducibility of your results, we recommend that if applicable you deposit your laboratory protocols in protocols.io, where a protocol can be assigned its own identifier (DOI) such that it can be cited independently in the future. For instructions see: http://journals.plos.org/plosone/s/submission-guidelines#loc-laboratory-protocols

We look forward to receiving your revised manuscript.

Kind regards,

Chongyi Wei, DrPH

Academic Editor

PLOS ONE

Reviewers' comments:

Reviewer's Responses to Questions

**Comments to the Author**

1. If the authors have adequately addressed your comments raised in a previous round of review and you feel that this manuscript is now acceptable for publication, you may indicate that here to bypass the “Comments to the Author” section, enter your conflict of interest statement in the “Confidential to Editor” section, and submit your "Accept" recommendation.

Reviewer #1: All comments have been addressed

Reviewer #2: All comments have been addressed

Reviewer #3: All comments have been addressed

2. Is the manuscript technically sound, and do the data support the conclusions?

Reviewer #1: Yes

Reviewer #2: Yes

Reviewer #3: Yes

3. Has the statistical analysis been performed appropriately and rigorously? 

Reviewer #1: Yes

Reviewer #2: Yes

Reviewer #3: Yes

4. Have the authors made all data underlying the findings in their manuscript fully available?

Reviewer #1: Yes

Reviewer #2: No

Reviewer #3: (No Response)

5. Is the manuscript presented in an intelligible fashion and written in standard English?

Reviewer #1: Yes

Reviewer #2: Yes

Reviewer #3: Yes

6. Review Comments to the Author

Reviewer #1: (No Response)

Reviewer #2: The authors have addressed most points satisfactorily, but there are a few outstanding issues.

1. The authors state: “The Chinese government has advocated for an increase in HIV

51 testing among high-risk populations since 2012.” However, it is clear that testing among high-risk populations was advocated much earlier than 2012. See http://data.unaids.org/una-docs/china_joint_assessment_2003_en.pdf

2. Line 135. Change “HIV men” to men living with HIV, or something else.

3. Lines 142-145. “Enrolled participants proceeded to the next round if tested…” What do you mean by “the next round if tested”? Please restructure/reword this paragraph, which I found to be somewhat confusing.

4. Line 234. “to identify a positive.” Please reword.

5. There are still typos/grammatical issues in the manuscript. For example, “sexual partner” on lines 243 and 245 and Table 2 (Frequency of sexual partner) should be changed to “sex”. And “mod” is misspelled on line 254.

6. Lines 291-292. “The mean number of successfully recruiting a reachable

292 sexual partner for HIV testing was 1.1.” Please rephrase for clarity.

7. Lines 329-331. “This may reflect the fact that the choice of information assisted mode may indicate a less intimate relationship.” Please elaborate on the connection between an intimate relationship and lower effectiveness in case finding. That connection may not be intuitive to the reader.

8. Grammar. Line 350. “However, the gap between total disclosed sexual partners (4116) and

351 those in recent six months (1014) is also need to be considered, as…”

9. Line 358. Given the stated study population, I don’t quite understand the issue with selection bias. The stated study population on lines 103-106 is MSM who agreed to participate in the program. This would suggest that MSM who did not agree to participate were NOT in the study population of interest. Perhaps the authors meant the study sample were MSM who tested positive and agreed to participate and that the study population was all MSM who tested positive?

10. The wording of the stated aim could be clearer. “In this study, we describe a pilot study of a choice-based partner tracing and testing program aimed at analyzing the uptake and infection status of HIV testing for sexual partners of newly diagnosed HIV-positive MSM.” Is it the study that is aiming to analyze the uptake and infection status? Or is it the program itself that aimed to analyze the uptake and infection status?

Recommendation: MINOR REVISION

Reviewer #3: Thank you for taking the time to make revisions on your manuscript. I believe that the paper now reads significantly better and conveys your findings in a clear way.

7. PLOS authors have the option to publish the peer review history of their article (what does this mean?). If published, this will include your full peer review and any attached files.

Reviewer #1: No

Reviewer #2: No

Reviewer #3: No

---

## [Author Response · Author response to Decision Letter 1]

10 Apr 2020

Dear editor and reviewers，

We appreciate a lot for your comments which are critical to help us improve this manuscript. We take these comments into account carefully, and have revised this manuscript accordingly. All authors of this manuscript would like to express our sincere thanks for your consideration of this resubmission.

 The reviewers’ comments are responded as follows point by point.

Reviewer #2: The authors have addressed most points satisfactorily, but there are a few outstanding issues.

1. The authors state: “The Chinese government has advocated for an increase in HIV

 testing among high-risk populations since 2012.” However, it is clear that testing among high-risk populations was advocated much earlier than 2012. See http://data.unaids.org/una-docs/china_joint_assessment_2003_en.pdf

Thanks a lot for your comment. We have checked this assessment and related polices，and read the document A Joint Assessment of HIV/AIDS Prevention, Treatment and Care in China provide by the reviewer. In 2003, Chinese government issued the national policy of "Four Free and One Care", which included free HIV counselling and testing (VCT), free antiviral treatment, free prevention of transmission from mother to child, free education for HIV affected children, and care for people living with HIV. We have revised this sentence into “In 2003, the Chinese government began to enforce the free voluntary counselling and testing (VCT) policy.” In lines 51-53. The relevant reference was also changed.

2. Line 135. Change “HIV men” to men living with HIV, or something else.

We have revised “HIV men” into “HIV positive MSM”, in lines 134.

3. Lines 142-145. “Enrolled participants proceeded to the next round if tested…” What do you mean by “the next round if tested”? Please restructure/reword this paragraph, which I found to be somewhat confusing.

This sentence has been revised as “Enrolled sexual partners proceeded to the next round if tested HIV positive and consented to be enrolled as ICs.” In lines 141-142.

4. Line 234. “to identify a positive.” Please reword.

We have revised the whole sentence into “Overall, the proportion of HIV testing among reachable sexual partner for HIV testing was 87%, and HIV testing yield among tested sexual partners was 16%.” In lines 231-233.

5. There are still typos/grammatical issues in the manuscript. For example, “sexual partner” on lines 243 and 245 and Table 2 (Frequency of sexual partner) should be changed to “sex”. And “mod” is misspelled on line 254.

We have checked and revised these grammatical or spelling mistakes.

6. Lines 291-292. “The mean number of successfully recruiting a reachable

292 sexual partner for HIV testing was 1.1.” Please rephrase for clarity.

We think this sentence may be a little redundant and not clear, and removed this sentence.

7. Lines 329-331. “This may reflect the fact that the choice of information assisted mode may indicate a less intimate relationship.” Please elaborate on the connection between an intimate relationship and lower effectiveness in case finding. That connection may not be intuitive to the reader.

This sentence has been revised in lines 329-330 to clarify that less intimate relationship may result in the difficulty for ICs to reach their sexual partners.

8. Grammar. Line 350. “However, the gap between total disclosed sexual partners (4116) and

351 those in recent six months (1014) is also need to be considered, as…”

This sentence has been revised as “However, the number of sexual partners in recent six months (1,014) was much less than the number of sexual partners in their whole life (4,116), which is also need to be considered, as…” in lines 349-350.

9. Line 358. Given the stated study population, I don’t quite understand the issue with selection bias. The stated study population on lines 103-106 is MSM who agreed to participate in the program. This would suggest that MSM who did not agree to participate were NOT in the study population of interest. Perhaps the authors meant the study sample were MSM who tested positive and agreed to participate and that the study population was all MSM who tested positive?

As descripted in lines 102-105, we defined the study population as newly diagnosed HIV-positive MSM and their sexual partners, who agreed to participate in the partner tracing and HIV testing program at VCT clinic study sites in the cities of Hangzhou and Ningbo from June 2014 through June 2016. We agree that it may not be suitable to name this as selection bias. This sentence is revised into “The demographic characteristics of HIV positive MSM might be associate with their participation in this pilot study” 357-359.

10. The wording of the stated aim could be clearer. “In this study, we describe a pilot study of a choice-based partner tracing and testing program aimed at analyzing the uptake and infection status of HIV testing for sexual partners of newly diagnosed HIV-positive MSM.” Is it the study that is aiming to analyze the uptake and infection status? Or is it the program itself that aimed to analyze the uptake and infection status?

This study aims to analyze the uptake and the infection status. We have revised this sentence into “In this study, we aimed at analyzing the uptake and infection status of HIV testing for sexual partners of newly diagnosed HIV-positive MSM.” In lines 95-97.

---

## [Editor Report · Decision Letter 2]

13 Apr 2020

HIV testing uptake and yield among sexual partners of HIV-positive men who have sex with men in Zhejiang Province, China, 2014-2016: a cross-sectional pilot study of a choice-based partner tracing and testing package

PONE-D-19-30003R2

Dear Dr. Pan,

We are pleased to inform you that your manuscript has been judged scientifically suitable for publication and will be formally accepted for publication once it complies with all outstanding technical requirements.

With kind regards,

Chongyi Wei, DrPH

Academic Editor

PLOS ONE
---

## [Editor Report · Acceptance letter]

24 Apr 2020

PONE-D-19-30003R2 

HIV testing uptake and yield among sexual partners of HIV-positive men who have sex with men in Zhejiang Province, China, 2014-2016: a cross-sectional pilot study of a choice-based partner tracing and testing package 

Dear Dr. Pan:

I am pleased to inform you that your manuscript has been deemed suitable for publication in PLOS ONE. Congratulations! Your manuscript is now with our production department. 

With kind regards,

on behalf of

Dr. Chongyi Wei 

Academic Editor

PLOS ONE